# Tanh Works Better With Asymmetry

**Dongjin Kim**[1,3] **Woojeong Kim**[2] **Suhyun Kim**[3]*

[1]Department of Computer Science and Engineering, Korea University
[2]Department of Computer Science, Cornell University
[3]Korea Institute of Science and Technology
{npclinic3, kwj962004, dr.suhyun.kim}@gmail.com

## Abstract

Batch Normalization is commonly located in front of activation functions, as proposed by the original paper. Swapping the order, i.e., using Batch Normalization after activation functions, has also been attempted, but its performance is generally not much different from the conventional order when ReLU or a similar activation function is used. However, in the case of bounded activation functions like Tanh, we discovered that the swapped order achieves considerably better performance than the conventional order on various benchmarks and architectures. This paper reports this remarkable phenomenon and closely examines what contributes to this performance improvement. By looking at the output distributions of individual activation functions, not the whole layers, we found that many of them are asymmetrically saturated. The experiments designed to induce a different degree of asymmetric saturation support the hypothesis that *asymmetric saturation helps improve performance*. In addition, Batch Normalization after bounded activation functions relocates the asymmetrically saturated output of activation functions near zero, enabling the swapped model to have high sparsity, further improving performance. Extensive experiments with Tanh, LeCun Tanh, and Softsign show that the swapped models achieve improved performance with a high degree of asymmetric saturation. Finally, based on this investigation, we test a Tanh function shifted to be asymmetric. This shifted Tanh function that is manipulated to have consistent asymmetry shows even higher accuracy than the original Tanh used in the swapped order, confirming the asymmetry's importance. The code is available at `https://github.com/hipros/tanh_works_better_with_asymmetry`.

## 1 Introduction

Batch Normalization (BN) is a widely used technique in deep learning. It was proposed to address the internal covariate shift problem by maintaining a stable output distribution among layers. As the characteristics of the output distribution of the weighted summation operation tend to be symmetric, non-sparse, and Gaussian [10], Ioffe & Szegedy [11] placed the BN between the weight and activation function. Thus, the "weight-BN-activation" order, which we call "Convention" in this paper, has been widely used to construct a block in many architectures [18, 8]. "Swap" models, swapping the order of BN and the activation function in a block, have also been attempted, but no significant and consistent difference between the two orders has been observed in the case of ReLU or similarly shaped activation functions. For instance, Hasani & Khotanlou [6] evaluated the effect of the position of BN in terms of training speed and concluded that there is no clear winner and the result depends on the datasets and architecture types.

---

*Corresponding author

37th Conference on Neural Information Processing Systems (NeurIPS 2023).

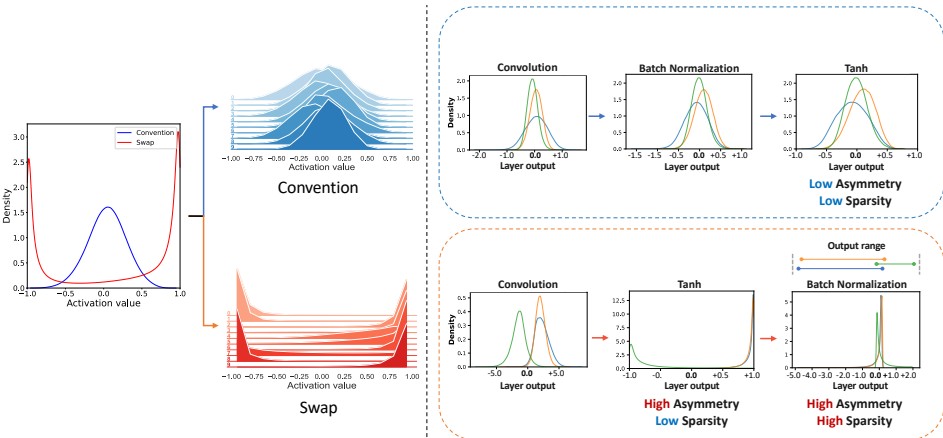

Figure 1: We examine the layer and channel-wise distributions of Tanh activation (left) and the feature map distribution in a single convolution block (right). Tanh produces nearly symmetrical layer activation in the Convention and Swap models (left-left). However, the channel distributions vary, with the Convention model showing symmetry (left top) and the Swap model presenting one-sided distributions (left bottom). The Convention model, with zero-centered Tanh input, exhibits low asymmetry and sparsity (right top). The Swap model, without the zero-centered constraint of BN, enhances asymmetry and, due to BN's placement after Tanh, sparsity (right bottom). The ranges of output are marked as lines above the plot.

However, in the case of bounded activation functions, we empirically found that the Swap order exhibits substantial improvements in test accuracy than the Convention order with diverse architectures and datasets. This paper empirically investigates the reason for this accuracy difference between the Convention and the Swap models with bounded activation functions. For simplicity, our analyses are mainly conducted with Tanh but are applicable to similar antisymmetric and bounded activation functions. We present the results with LeCun Tanh and Softsign at the end of the experimental section.

One key difference between Swap and Convention models is the distribution of activation values, as shown in Figure 1(left). In the Swap model, the mean of convolution layer output shifts substantially away from 0. Thus, most activation values are near the asymptotic values of the bounded activation function, that is, highly saturated. This is unanticipated since it is a common belief that high saturation should be avoided. To investigate this paradox, we took one step further and looked at the output distribution of each activation function, not just a whole layer. To our very surprise, even though the distribution is fairly symmetric at the layer level, the activation values of each channel are biased toward either one of the asymptotic values or *asymmetrically saturated*. Through an in-depth investigation, we come to a hypothesis that this asymmetric saturation is a key factor for the performance improvement of the Swap model since it enables Tanh to behave like a one-sided activation function. In the experiments to examine whether asymmetric saturation is related to the performance of models, we observe that the accuracy and the degree of asymmetric saturation are highly correlated.

BN after Tanh also shifts the distribution biased toward either -1 or 1 by asymmetric saturation near zero, which has the important effect of increasing sparsity, as depicted in Figure 1(right). Sparsity is widely considered to be a desirable property. For instance, Glorot et al. [5] studied the benefits of ReLU compared to Tanh in terms of sparsity. One thing to note is that if each channel is symmetrically saturated, BN will not increase sparsity much since the mean is already close to zero. In contrast, the one-sided property of asymmetric saturation causes at least half of the sample values after normalization to be almost zero. It allows the Swap model to have even higher sparsity than the Convention model. Asymmetric saturation combined with normalization makes a bounded activation function behave much like ReLU.

Based on the above lessons, we test enforcing the asymmetric saturation and sparsity of Tanh outputs. This involves shifting the inputs of Tanh by $\tau$, introducing asymmetry, and then shifting the outputs in reverse by $-tanh(\tau)$ to boost sparsity. This is equivalent to using a shifted Tanh as a new activation function. Our experimentation across different architectures and datasets reveals that this adjusted

Tanh function with fixed asymmetry consistently outperforms the original Tanh used in the Swap order. Its effectiveness is even comparable to that of ReLU. This validates the hypothesis that Tanh works more efficiently with asymmetric saturation.

Our findings are as follows:

- The Swap model using Batch Normalization after bounded activation functions performs better than the Convention model in many architectures and datasets.

- We discover the *asymmetric saturation* at the channel level and investigate its importance through carefully designed experiments.

- We identify the high sparsity induced by Batch Normalization after bounded activation functions and perform an experiment to examine the impact of sparsity on performance.

- Our shifted Tanh with enforced asymmetric saturation shows higher accuracy than the Swap Tanh, which reconfirms the importance of asymmetry.

## 2  Settings and Notation

**Model**  For our investigation, we choose the VGG-like model due to its simple and straightforward design. This model is trained on the CIFAR-100 dataset [12]. As a modification, we replace the ReLU activation function with Tanh. However, because the VGG architecture was proposed for the ImageNet dataset, the model is overparameterized for the CIFAR dataset. It incurs poor performance and difficulty in investigating the Swap model. Thus, we cut out the last convolution layers and select the best model based on the validation accuracy. The model with five cut-out layers shows the best accuracy, as in the Appendix. We call this model "VGG16_11" and use this architecture to investigate Convention and Swap orders.

**Metric**  We consider three properties to investigate each order: saturation, asymmetric saturation, and sparsity. Saturation is measured on Tanh and quantifies how closely outputs the values to the maximum absolute output range limit, also known as the asymptotic value. We involve the skewness metric in statistics to measure asymmetry on Tanh. One thing that differs from the skewness is the absolute. We absolute the skewness to quantify the asymmetry regardless of skewed direction. Sparsity is measured on the output of the block, i.e., Tanh for Convention and BN for Swap. To measure the sparsity, we inversely utilize our saturation metric.

**Experiment setup**  In Section 4.2, we verify the effect of asymmetry by slightly cutting off the asymmetry in the asymmetry-boosted Convention model, which we refer to the No-Weight-Decay-on-BN (NWDBN) model. The NWDBN model is ordered as Convention, and the weight decay does not apply to BN. We fix the intensity of the weight decay on the weight layer and vary the decay intensity on BN to regulate asymmetry. The initial hyperparameters followed the best hyperparameter of the NWDBN model. Then, we increase the intensity of weight decay on $\beta$ in BN from 0.0 to 0.001 by 0.0001. In Section 5.3, we investigate the effect of sparsity by slightly increasing it. The initial hyperparameters followed the Swap model's best hyperparameters. Then, we change the weight decay intensity on the $\gamma$ and $\beta$ in BN. The searched intensities of weight decay are 0, 1e-6, 5e-6, 1e-5, 5e-5, 1e-4, and 5e-4. We verify the effects of the layer order in various datasets, networks, and activation functions in Section 7.1. We train models on four benchmarks (CIFAR-10, CIFAR-100, Tiny ImageNet [13], and ImageNet [3]), three base-architectures (VGG, MobileNet, PreAct-ResNet), and three activation functions (ReLU, Tanh, ShiftedTanh). For the ResNet, the Swap order incurs extra non-linearity functions, which hinder the fair comparison between the Convention and Swap model. Except for the first stage, the first shortcut connection in each stage has convolution and BN layers in the block. In the case of the Swap Tanh model, Tanh replaces BN and vice-versa. Thus, the number of non-linearity functions increases as the number of the linear projection layer in the shortcut. We, therefore, report the results of PreAct-ResNet, which only has the convolution layer in the shortcut connection. (Furthermore, upon training ResNet20 with Tanh on the CIFAR-100 dataset, we observed that the Swap model surpassed the Convention model, achieving an accuracy of 69.06% compared to the Convention model's 68.97%.)

All results, except the ImageNet dataset, are conducted on three random seeds. The measured values and accuracy are averaged over seeds. We use the SGD optimizer with a momentum of 0.9, weight

decay. We adopt a 2-step learning rate decaying strategy that decays by 0.1. We conduct a grid search to obtain the best model for investigation. We search different combinations of initial learning rates (0.1 and 0.01) and weight decay values (1e-4, 5e-4, 1e-3, and 5e-3). The specific hyperparameters for each model can be found in the Appendix.

**Notation** The input of an arbitrary layer in the convolution neural network is denoted as $\mathcal{X} \in \mathbb{R}^{n \times c \times h \times w}$, where $n$, $c$, $h$, and $w$ represent the number of samples, channels, height, and width, respectively. We reshape $\mathcal{X}$ to $X \in \mathbb{R}^{c \times m}$, where $m = nhw$. The output of the Tanh function is $T \in \mathbb{R}^{c \times m}$ where $T_{i,j} = tanh(X_{i,j})$. BN normalizes the channel distribution to have zero mean and unit variance using estimated statistics. After normalization, an affine transformation with scaling parameter $\gamma \in \mathbb{R}^c$ and shifting parameter $\beta \in \mathbb{R}^c$ is applied channel-wise. Thus, the output of BN is given by: $B_{i,j} = \gamma_i \left( \frac{X_{i,j} - \hat{\mu}_i}{\sqrt{\hat{\sigma}_i^2 + \epsilon}} \right) + \beta_i$, where $\epsilon$ is a constant added for numerical stability, $\hat{\mu}_i$ and $\hat{\sigma}_i$ are estimated mean and standard deviation of each channel, respectively, computed over the mini-batch of training data samples.

## 3    Overly Saturated but Well-Generalized Model

The phenomenon of saturation refers to the state in which a neuron predominantly outputs values close to the asymptotic ends of the bounded activation function [15]. When training a neural network with bounded activation functions, the outputs of neurons increase due to the gradually increasing weights [4]. The increased output values become close to the near asymptote in bounded activation functions, as shown in the experiment in Glorot & Bengio [4]. However, excessive saturation damages the neural network, and it is regarded as a situation to evade [4, 15, 16, 1, 2, 11]. Various methods were proposed to prevent excessive saturation. Glorot & Bengio [4] proposed an initialization scheme, Rakitianskaia & Engelbrecht [15, 16] proposed a metric to measure the degree of saturation for monitoring the training, Bhat et al. [1] pre-scaled the inputs of the activation function, and Chen & Chang [2] proposed adaptable bounded activation.

### 3.1    Saturation Metric

Our saturation metric quantifies the extent to which the output values approach the function's output range limit, also known as the asymptotic value. To normalize the input, we calculate the absolute value and divide it by the maximum absolute value, resulting in a range of [0, 1]. The saturation $t$ of a layer is defined as:

$$t = \frac{1}{c} \Sigma_{i=1}^c \frac{\Sigma_{j=1}^m |T_{i,j}|}{ma}, \tag{1}$$

where $a$ represents the maximum absolute value of the output range of the activation function. Specifically, for the Tanh function, $a$ is equal to 1. $t$ lies in $[0, 1)$. The large value of this metric indicates $T$ are highly saturated, as illustrated in the Appendix.

### 3.2    High Saturation in the Swap Model

Even if only the layer order is changed from the Convention order to the Swap order, there is a 4.61% improvement in test accuracy. However, when we measure the layer saturation in both models, the Swap model has highly saturated layers. The layer-wise Saturation of Swap and Convention models can be seen in Figure 2(a). This is counterintuitive as excessive saturation is considered undesirable in previous works [4, 15, 16, 1, 2]. The saturation of almost half of the layers in the Swap model is over 0.7. Some layers even have a saturation over 0.8. On the other hand, the saturation of the Convention model is lower than 0.5 over all layers.

## 4    Asymmetric Saturation

In order to scrutinize saturation, we perform an in-depth analysis at the channel level, one step further than the layer level. Fascinatingly, we notice that most channels display asymmetric saturation in the Swap model, especially in layers with excessive saturation. On the other hand, the channel distribution in the Convention model tends to be centered around zero, even at the channel level.

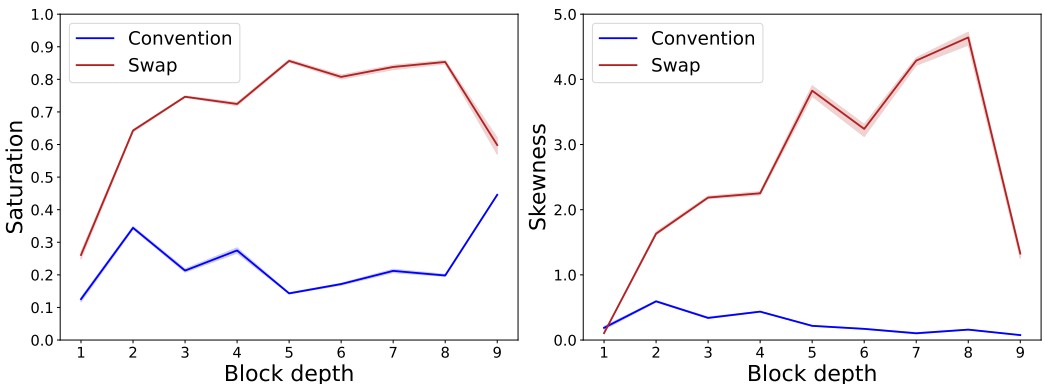

Figure 2: (a) Layer-wise Saturation of Convention and Swap models. (b) Layer Skewness in Convention and Swap models. The two figures display a 95% confidence interval.

## 4.1 Asymmetric Saturation Metric

Both the Convention and Swap models demonstrate symmetric distributions at the layer level. However, notable differences arise at the channel level. To quantify and compare the degree of asymmetry present at the channel level, we measure the asymmetry of each channel individually and calculate the average to determine the overall skewness of the layer. We adopt sample skewness as a metric to gauge the asymmetry of the Tanh output. The skewness of a given $i^{\text{th}}$ channel, denoted $p_i$, is calculated as follows:

$$p_i = \frac{m^2}{(m-1)(m-2)} \frac{\mu_i^{\beta}}{\sigma_i^3} \tag{2}$$

where $\mu_i^{\beta}$ represents the sample third central moment of $T_{i,:}$, while $\sigma_i^3$ represents the sample standard deviation cubed of $T_{i,:}$. The averaged absolute skewness of a layer, $k$, herein referred to simply as Skewness, is given by:

$$k = \frac{1}{c} \Sigma_{i=1}^c |p_i|. \tag{3}$$

Figure 2(b) illustrates the Skewness for all layers in the Convention and Swap models. In the Convention model, the Skewness of most layers is close to zero, indicating a minimal presence of asymmetric distribution. Conversely, the Swap model exhibits higher Skewness compared to the Convention model. Moreover, the regions with high saturation blocks within the Swap model show particularly elevated Skewness values. This suggests that saturation in these areas occurs asymmetrically in the channel distributions.

## 4.2 Effect of Asymmetric Saturation on Performance

In order to examine the effect of asymmetric saturation, we devise a method to control the level of asymmetry in the Convention model. One key factor that limits the emergence of asymmetric saturation in the Convention model is the influence of weight decay on $\gamma$ and $\beta$ within BN. Especially, small $\beta$ values by weight decay constrain the distribution to be zero-centered, which incurs a decrease in the asymmetry of the Tanh output. In our experiment, the NWDBN model successfully encourages asymmetric saturation and has higher Skewness than the Convention model. Moreover, the NWDBN model shows improved accuracy of **72.22%** compared to the Convention model **69.5%**. To better examine the effects of asymmetric saturation on performance, we gradually increase the intensity of weight

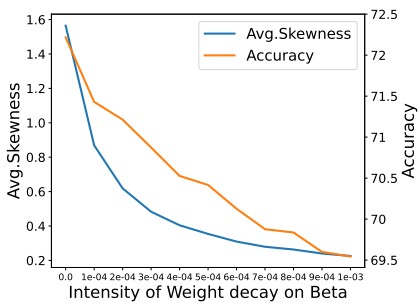

Figure 3: Relation between accuracy and averaged Skewness over layers. The "Avg. Skewness" averaged all the layer-wise Skewness in each model.

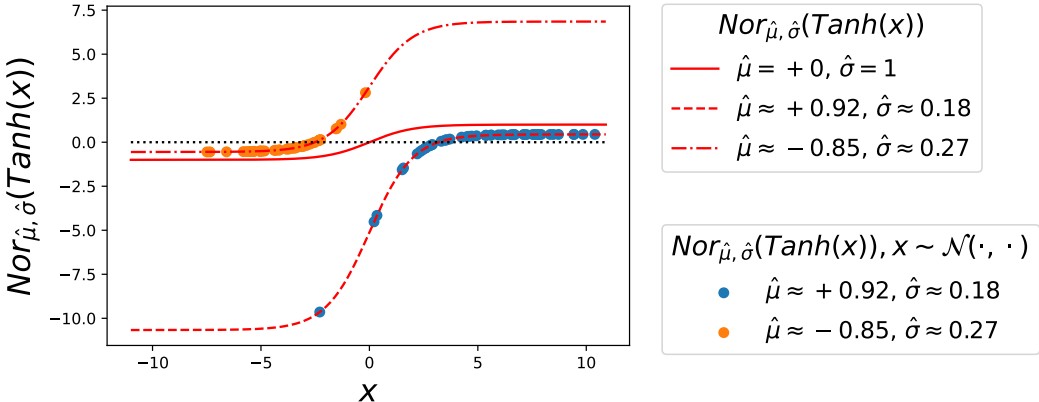

Figure 4: Red lines represent Tanh activation functions combined with normalization. The dots represent the output values of the combined function for different input samples. Samples of the same color indicate they generate the same BN statistics on the Tanh output. The notation $Nor_{\hat{\mu},\hat{\sigma}}$ represents the normalization function of BN.

decay on the $\beta$ parameter from zero, which indicates that slightly eliminates the asymmetry in the NWDBN model. As the weight decay on $\beta$ gets bigger, Skewness becomes smaller, and the test accuracy degrades, as shown in Figure 3. This strong correlation clearly supports our hypothesis: *asymmetric saturation helps improve performance.*

## 5 Sparsity

### 5.1 Asymmetric Saturation with Batch Normalization Can Induce High Sparsity

ReLU achieves a high generalization performance by utilizing the strengths of sparsity [5]. The sparsity introduced by ReLU is a consequence of its single boundary at zero, which transforms all negative inputs to zero. In addition, work by Ramachandran et al. [17] shows the benefits of having a single asymptote at zero. They carried out an automated strategy to investigate a variety of activation functions. The most notable activation functions discovered through this search were one-sided and had a boundary value close to zero, similar to ReLU.

We note that the Swap model effectively generates sparsity, pushing a majority of values toward zero when asymmetric saturation occurs. This shifting is credited to the normalization process in Batch Normalization (BN), which recenters the distribution around zero. Thus, when the asymmetric saturation occurs, this saturation is subsequently shifted towards zero during the normalization process, enhancing sparsity. The combined effect of Tanh with normalization, along with samples of the function's output showing different BN statistics in the Swap order, are depicted in Figure 4. The majority of values and the asymptotic lines that the samples saturated are shifted to near zero.

### 5.2 Sparsity Metric

Intuitively, a sparse representation is one in which a small number of coefficients contain a large proportion of the energy [9]. We utilize our saturation metric to measure sparsity. Saturation quantifies the extent to which many values reach the maximum. We measure the sparsity by the extent to which many values are far from the maximum. However, as sparsity is measured on the block output, we consider the output of Tanh for the Convention model and the output of BN for the Swap model. BN has no boundaries, so we empirically calculate $a$ for normalization in Equation 1. We denote the empirical saturation, $\hat{t}$, as follows:

$$\hat{t} = \frac{1}{c}\Sigma_{i=1}^{c}\frac{\Sigma_{j=1}^{m}|B_{i,j}|}{mB_{i,:}^{\max+}}, \tag{4}$$

where $B_{i,:}^{max}$ is the maximum absolute element in $B_{i,:}$, i.e., $B_{i,:}^{max+} = \max_j B_{i,j}$. For the Convention model, we apply the above metric to $T$ instead of $B$. Finally, the sparsity $s$ is defined as:

$$s = 1 - \hat{t} \tag{5}$$

Our sparsity metric satisfies five criteria among six desired criteria of sparsity measures in Hurley & Rickard [9]. The proof of each criterion can be found in the Appendix.

### 5.3 Effect of Sparsity on Performance

In this section, we investigate the significance of sparsity in the Swap model and its influence on test accuracy. There's a marked relationship between increased sparsity and test accuracy. The result can be seen in Figure 5.

As elaborated in Section 5.1, the Swap order has the capability to intensify sparsity when there's asymmetric saturation. However, the introduced sparsity in the Swap model might be reduced due to the shifting operation present in the affine transformation of BN. This reduction occurs because effectively decaying the affine parameters can help maintain values close to zero throughout the training. To preserve the sparsity in the

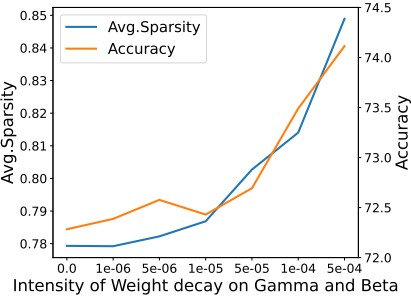

Figure 5: Relation between accuracy and averaged Sparsity over layers. The "Avg. Sparsity" averaged all the layer-wise Sparsity in each model.

Swap model, we increase the weight decay on the parameters involved in the affine transformation. Such an amplified weight decay can further boost the sparsity of the BN output.

## 6 Shifted Tanh for Asymmetric and Sparse Output

Based on the investigation, we modify the Tanh function, referred to as Shifted Tanh, to encourage asymmetry and sparsity. Different from the Swap Tanh model, Shifted Tanh can generate asymmetry and sparse activation on zero-centered input. Shifted Tanh shares properties similar to ReLU. This property accelerates the model to use asymmetry and sparsity for better generalization.

The Shifted Tanh involves shifting the inputs by a factor of $\tau \in \mathbb{R}$ before the Tanh operation, introducing asymmetry, and then shifting the outputs in reverse by the degree of the shift while considering the Tanh operation. The formulation of the Shifted Tanh is as follows:

$$x \mapsto tanh(x + \tau) - tanh(\tau). \tag{6}$$

We choose $\tau = -1$ based on the best accuracy from the ablation study on weight $\tau$, as shown in the Appendix.

## 7 Extended Experiments

### 7.1 Investigation on Various Datasets and Architectures

Our study delves into the Tanh models trained on the CIFAR-100 dataset using the VGG16_11 architecture. In this section, we evaluate the Swap order in various datasets and architectures and compare its accuracy against the Convention order.

From the results presented in Table 1, a significant difference is observed in performance between the Swap and Convention orders when using Tanh; the Swap order consistently delivers better results. In contrast, for both ReLU and shifted Tanh models, the sequence of layers has a negligible effect on accuracy. Moreover, the Convention model usually exhibits a slight edge over the Swap model. We assume that the zero-centered distribution of BN augments the asymmetry and sparsity of the Shifted Tanh in the Convention arrangement. Hence, the Convention model's accuracy slightly surpasses the Swap model, mirroring patterns seen with ReLU. It's noteworthy that the enhanced accuracy of shifted Tanh rivals that of the ReLU model.

Table 1: Experimental comparison between orders on various datasets and models.

| Datasets | Models | Tanh | | Shifted Tanh | | ReLU | |
|---|---|---|---|---|---|---|---|
| | | Convention | Swap | Convention | Swap | Convention | Swap |
| CIFAR-10 | VGG16 | $91.75_{\pm 0.08}$ | $92.90_{\pm 0.05}$ | $93.92_{\pm 0.17}$ | $93.60_{\pm 0.17}$ | $93.69_{\pm 0.08}$ | $93.04_{\pm 0.11}$ |
| | MobileNet | $91.66_{\pm 0.03}$ | $92.53_{\pm 0.12}$ | $92.85_{\pm 0.10}$ | $92.64_{\pm 0.21}$ | $92.48_{\pm 0.23}$ | $91.93_{\pm 0.30}$ |
| | PreAct-ResNet18 | $92.46_{\pm 0.17}$ | $94.41_{\pm 0.15}$ | $94.48_{\pm 0.15}$ | $94.57_{\pm 0.18}$ | $94.94_{\pm 0.07}$ | $94.86_{\pm 0.18}$ |
| CIFAR-100 | VGG16 | $64.95_{\pm 0.31}$ | $70.93_{\pm 0.12}$ | $73.87_{\pm 0.05}$ | $73.21_{\pm 0.31}$ | $73.68_{\pm 0.08}$ | $71.79_{\pm 0.09}$ |
| | MobileNet | $64.50_{\pm 0.34}$ | $70.39_{\pm 0.13}$ | $72.75_{\pm 0.22}$ | $71.88_{\pm 0.54}$ | $70.27_{\pm 0.22}$ | $69.49_{\pm 0.28}$ |
| | PreAct-ResNet18 | $73.26_{\pm 0.19}$ | $75.76_{\pm 0.17}$ | $76.70_{\pm 0.35}$ | $76.51_{\pm 0.21}$ | $78.06_{\pm 0.29}$ | $77.39_{\pm 0.23}$ |
| Tiny ImageNet | VGG16 | $49.29_{\pm 0.18}$ | $57.05_{\pm 0.28}$ | $60.68_{\pm 0.09}$ | $60.37_{\pm 0.07}$ | $59.37_{\pm 0.09}$ | $59.05_{\pm 0.33}$ |
| | MobileNet | $45.27_{\pm 0.19}$ | $51.78_{\pm 0.26}$ | $52.91_{\pm 0.26}$ | $52.86_{\pm 0.15}$ | $51.92_{\pm 0.11}$ | $50.48_{\pm 0.26}$ |
| | PreAct-ResNet34 | $59.06_{\pm 0.26}$ | $64.94_{\pm 0.05}$ | $65.82_{\pm 0.18}$ | $65.66_{\pm 0.20}$ | $67.28_{\pm 0.23}$ | $66.21_{\pm 0.18}$ |
| ImageNet | VGG16 | 60.85 | 67.04 | 73.4 | 72.11 | 73.83 | 72.95 |
| | MobileNet | 64.26 | 72.07 | 70.4 | 72.56 | 68.27 | 71.1 |

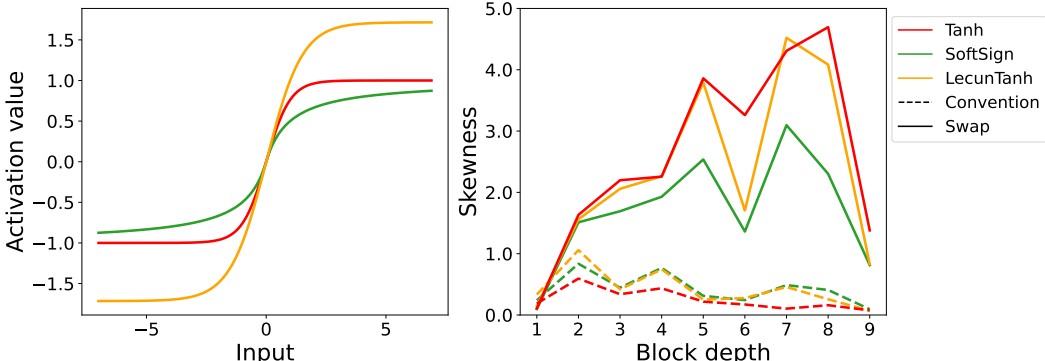

Figure 6: Shapes of activation functions (left) and Skewness tendency of different activation functions (right). The dashed line represents the Convention model and the solid line represents the Swap model in the right figure.

## 7.2 Effect of the Swap Order on Other Bounded Activation Functions

Asymmetric saturation is a crucial factor in the success of the Tanh model. The primary focus of our investigations lies with the Tanh activation function. However, in this section, we extend our examination to include other bounded activation functions, namely LeCun Tanh [14] and Softsign [19].

Softsign, defined as $\frac{x}{|x|+1}$, has the same asymptotic values as Tanh, yet its growth rate is polynomial, not exponential, resulting in a slower approach to its asymptotes [4]. Therefore, it can mitigate the issue of vanishing gradients by reducing neuron saturation. LeCun Tanh, characterized by a gentler slope and a broader output range than Tanh, has the formula $1.7159 \times Tanh(\frac{2}{3} \times x)$.

The asymmetric saturation caused by the Swap order occurs not only in Tanh but also in other bounded activation functions. The Convention model with the three types of bounded activation functions shows low layer-wise Skewness. The Skewness is less than one over the overall layer. On the other hand, a significant Skewness increment arises when the Swap order is applied. The shapes of these functions and layer Skewness are shown in Figure 6.

In all of the activation functions, the Swap models improve performance compared to the Convention. It can be found in Table 2. When swapping, asymmetric saturation happens the least in Softsign, which makes it challenging to create a saturation state. Furthermore, the Softsign model with the Swap order shows lower performance than the Tanh model with the Swap order, which could generate more saturation, even though the Convention model had the highest performance.

Table 2: Experimental comparison of accuracy with asymmetry between orders. We compare the accuracy with Skewness on VGG16_11 models trained by CIFAR-100. We used averaged Skewness over layers to calculate the difference of Skewness.

| Activation functions | Order | | Δavg. Skewness |
| --- | --- | --- | --- |
| | Convention | Swap | (Swap - Convention) |
| Tanh | 69.5 | 74.11 | 2.38 |
| Softsign | 70.01 | 73.65 | 1.28 |
| LeCun Tanh | 67.82 | 74.46 | 1.90 |

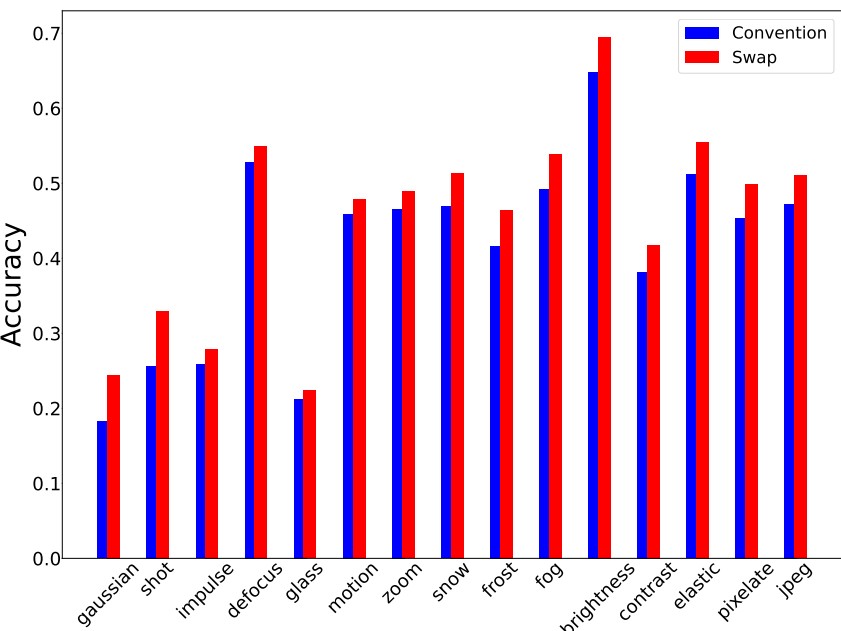

Figure 7: Experimental comparison between orders on the different types of corruption. We compare robust accuracy on CIFAR-100-C between the Convention and Swap ordered VGG16_11 trained by CIFAR-100.

## 7.3 Robustness of the Asymmetrically Saturated Model

We evaluate the Swap model using a corrupted version of the CIFAR-100 dataset, termed as CIFAR-100-C [7]. The results of robustness accuracy on CIFAR-100-C can be seen in Figure 7. The findings reveal an enhanced accuracy with the Swap order throughout all corruption categories. The Convention model has an average accuracy across all corruptions at **41.4%**. However, the Swap model enhances this average to **45.3%**.

Asymmetric saturation offers increased robustness to variations in input data. This robustness stems from the behavior of the Tanh function, especially when utilizing the output on its bounds to its midpoint. Near the center, particularly around zero, the Tanh function is highly responsive to slight input changes, leading to considerable alterations in the output. As a result, the Convention model is more vulnerable to such input shifts, especially in the zero vicinity of the Tanh function. In contrast, the Swap model activates Tanh's asymptotic region. Therefore, even pronounced input deviations lead to only slight output changes. Such a trait makes the Swap model inherently robust, maintaining a more consistent output even with input disparities.

Table 3: Experimental result of comparison between Convention and NWDBN models on VGG16_11 trained by CIFAR-100.

| Configurations | Accuracy | Avg. Skewness | Avg. Sparsity |
|---|---|---|---|
| Convention | 69.5 | 0.254 | **0.718** |
| NWDBN | 72.22 | **0.718** | 0.288 |

### 7.4 Further Discussion on the Effect of Asymmetry and Sparsity

The Swap model enhances its performance through both asymmetry and sparsity, making it challenging to distinguish the contributions of each. We choose the Convention order to determine the primary factor influencing performance. When asymmetry increases in the Convention model, the majority of values have large values and it decreases in sparsity. Thus, the NWDBN model, which encourages greater asymmetry in the Convention model, has a reduced sparsity while the level of asymmetry is enhanced. The result of accuracy and averaged Skewness and Sparsity on the Convention and NWDBN models can be seen in Table 3. Notably, even with its diminished sparsity, the asymmetrically saturated NWDBN model surpasses the Convention model's performance. From these observations, we deduce that asymmetry plays a more pivotal role than sparsity.

## 8 Conclusion

In this work, we report that the Swap models with bounded activation functions perform better than the Convention models and analyze what brings about performance improvement. Our analysis reveals that the asymmetric saturation and the increased sparsity by the batch normalization after bounded activation functions contribute to the increased accuracy of the Swap models. Interestingly, this makes bounded activation functions behave much like ReLU and enables them to achieve performance comparable to that of ReLU. Most importantly, it expands our understanding of the activation function and will further assist in developing superior activation functions in the future.

## 9 Acknowledgement

This work was supported by Institute of Information & Communications Technology Planning & Evaluation (IITP) grant funded by the Korea government (MSIT) (No.2021-0-00456, Development of Ultra-high Speech Quality Technology for Remote Multi-speaker Conference System), and by the National Research Council of Science & Technology (NST) grant by the Korea government (MSIT) [CRC-20-02-KIST].

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

# A    Limitation

While this paper contains an in-depth analysis of Tanh, it does not propose a function that is superior to existing activation functions. Generally, it is known that Tanh does not perform better than ReLU, but it's hard to find a study that deeply analyzes the reasons for this. To our knowledge, this paper is the most comprehensive analysis of the performance degradation when Tanh is used in a conventional manner. Although it suggests strategies to utilize Tanh more effectively, achieving performance nearly on par with ReLU, it fundamentally does not assert that Tanh surpasses ReLU in terms of effectiveness.

# B    Accuracy of the Shifted Tanh According to $\tau$

In this section, we focus on the hyperparameter $\tau$ in the shifted Tanh function. We carry out experiments using VGG16_11, MobileNet, and PreAct-ResNet models trained on CIFAR-10 and CIFAR-100 datasets, testing different values of the parameter $\tau$ (-1.5, -1.2, -1.0, -0.8, and -0.5). The performance results corresponding to each of these $\tau$ values are presented in Table B.1. In these experiments, all hyperparameters, except for $\tau$, are maintained at their best-averaged accuracy settings based on each model trained on the CIFAR dataset. Our aim is to identify the effective $\tau$ value without pushing all inputs into the saturation zone due to excessive asymmetry. The best-performance $\tau$ is identified by averaging the accuracy across different models for each $\tau$ and selecting the one with the highest performance. In our case, -1 is the best.

Table B.1: The results of shifted Tanh models on various $\tau$ values.

| $\tau$ | $-1.5$ | -1.2 | -1.0 | -0.8 | -0.5 |
|---|---|---|---|---|---|
| Accuracy | 80.56 | 83.86 | **84.15** | 84.01 | 83.58 |

# C    Metric

## C.1    Saturation and Skewness on Different Distribution Patterns

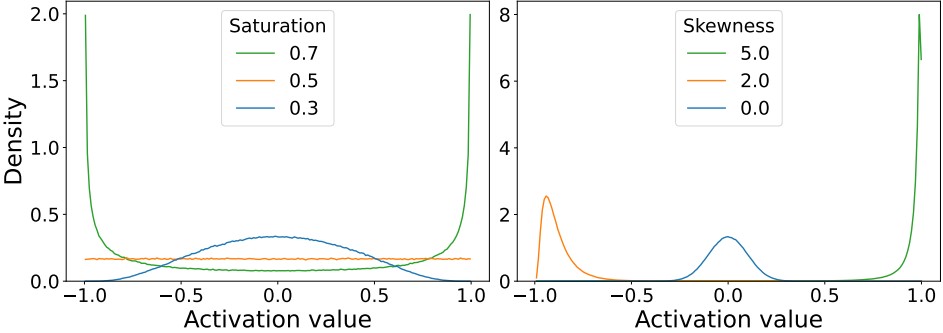

Figure C.1: The Saturation (left) and Skewness (right) values on different distributions.

Our saturation metric is designed to quantify the concentration of elements to the maximum absolute output values. The metric has bounds from 0 to 1, and it equates to 0 when all the elements are 0 in the Tanh case. Conversely, it grows larger as elements tend towards the maximum limit of the output range. For example, the saturation metric is evaluated at 0.5 in a uniform distribution. For saturation values of the Tanh output on the different Gaussian distributions, refer to Figure C.1 (left).

Skewness is a measure that quantifies the asymmetry within a distribution. Symmetric distribution will yield the Skewness of 0, and the Skewness increases with the rise in asymmetry. Notably, we calculate the absolute skewness value in our asymmetry metric, ensuring that the metric remains

unaffected by the direction of skewness. For a better understanding of how this measure applies to various distributions, Figure C.1 (right) provides Skewness on each distribution.

## C.2 Impacts of Mean and Standard Deviation Variations on Tanh Asymmetry

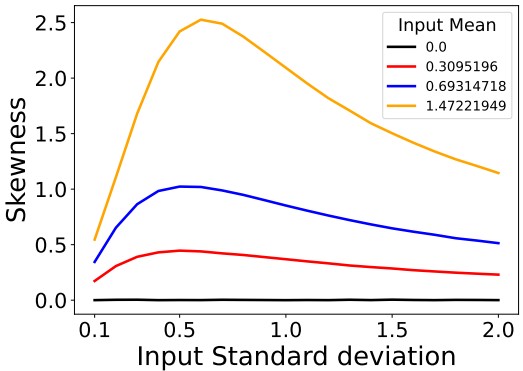

Figure C.2: Skewness value on the different means and standard deviations of the Gaussian distribution for Tanh input.

The mean and variance of input distribution on Tanh affect the asymmetry of Tanh output. The Skewness of Tanh output on various mean and standard deviation can be found in Figure C.2. In the Skewness of varied mean distribution, the Skewness is increased on the mean increase. However, the maximum Skewness does not align with the standard deviation increases. The Skewness decreases not only the small input standard deviation but also the large input standard deviation. Additionally, in the same mean condition, a decrease in standard deviation from the maximum skewness point more rapidly decreases the Skewness than an increase in standard deviation.

## C.3 The Proof of Demonstrating the Sparsity Metric Criteria

This section examines the adequacy of our sparsity metric against the heuristic criteria for sparsity measures defined by Hurley & Rickard [9]. Our sparsity metric meets five of these criteria: Scaling, Rising Tide, Cloning, Bill Gates, and Babies.

We consider the absolute operation as pre-applied to the metric input, according to Hurley & Rickard [9]. Thus, our sparsity metric is represented as $f : \mathbf{x} \mapsto s$, where $\mathbf{x} \in \mathbb{R}^{m+}$ is a vector of absolute values in a channel in BN output, and $x^{\mathrm{max}+}$ signifies the maximum value in $\mathbf{x}$, i.e., $x^{\mathrm{max}+} = \max_i x_i$. We prove our sparsity metric in the setting of one-channel, which can be expanded to the multi-channel setting without loss of generality. Below are the five criteria our sparsity metric meets and proof for each.

**Proof of Scaling.** Sparsity is scale invariant. That is, for any scalar $\alpha \in \mathbb{R}$, where $\alpha > 0$, the function $f$ satisfies:
$$f(\alpha \mathbf{x}) = f(\mathbf{x}).$$

The invariance of scale means that sparsity considers relative differences rather than absolute magnitudes.

*Proof.* If we scale the vector $\mathbf{x}$ by $\alpha$, we also scale the maximum positive value $x^{\mathrm{max}+}$ by $\alpha$. Therefore, we have:

$$
\begin{aligned}
f(\alpha \mathbf{x}) &= 1 - \frac{\sum_{i=1}^m \alpha x_i}{m \alpha x^{\mathrm{max}+}} \\
&= 1 - \frac{\sum_{i=1}^m x_i}{m x^{\mathrm{max}+}} \\
&= f(\mathbf{x}).
\end{aligned}
$$

This demonstrates that the sparsity of the vector $\mathbf{x}$ remains the same when the vector is scaled by any positive scalar $\alpha$. □

**Proof of Rising Tide.** Adding a constant to each element decreases sparsity. Formally, for any scalar $\alpha \in \mathbb{R}$, where $\alpha > 0$, the function $f$ satisfies:

$$f(\alpha + \mathbf{x}) < f(\mathbf{x}).$$

We exclude the case, as mentioned in [9], where all elements of $\mathbf{x}$ are the same.

This property also indicates that sparsity increases as more values approach zero.

*Proof.* We begin by noting the sparsity of the vector $\mathbf{x} + \alpha$:

$$f(\mathbf{x} + \alpha) = 1 - \frac{\Sigma_{i=1}^m x_i + m\alpha}{mx^{\mathrm{max}+} + m\alpha}.$$

Assume for contradiction that $f(\mathbf{x} + \alpha) \geq f(\mathbf{x})$. Given that $mx^{\mathrm{max}+} > \Sigma_{i=1}^m x_i > 0$ and $m\alpha > 0$ the following inequalities can be derived:

$$\frac{\Sigma_{i=1}^m x_i + m\alpha}{mx^{\mathrm{max}+} + m\alpha} > \frac{\Sigma_{i=1}^m x_i}{mx^{\mathrm{max}+}}$$

$$1 - \frac{\Sigma_{i=1}^m x_i + m\alpha}{mx^{\mathrm{max}+} + m\alpha} < 1 - \frac{\Sigma_{i=1}^m x_i}{mx^{\mathrm{max}+}}$$

which contradicts the initial assumption, thus implying that

$$f(\alpha + \mathbf{x}) < f(\mathbf{x}),$$

as desired. □

**Proof of Cloning.** Sparsity is invariant under cloning.

That is, for a vector $\mathbf{x}$, the function $f$ satisfies:

$$f(\mathbf{x}) = f(\mathbf{x}||\mathbf{x}) = f(\mathbf{x}||\mathbf{x}||\mathbf{x}) = \ldots = f(\mathbf{x}||\mathbf{x}||\ldots||\mathbf{x}),$$

where $||$ denotes concatenation, such that $\mathbf{x}||\mathbf{x} = [x_1, x_2, \ldots, x_m, x_1, x_2, \ldots, x_m]$.

The principle of cloning implies that sparsity remains the same even if a set of values is replicated.

*Proof.* Define a function $g : \mathbf{x}, \delta \mapsto \mathbf{y}$, where $\delta \in \mathbb{Z}^+$ and $\mathbf{y} \in \mathbb{R}^{m\delta}$ is the result of concatenating $\mathbf{x}$ $\delta$ times. We then have:

$$
\begin{aligned}
f(g(\mathbf{x}, \delta)) &= 1 - \frac{\delta\Sigma_{i=1}^m x_i}{\delta mx^{\mathrm{max}+}} \\
&= 1 - \frac{\Sigma_{i=1}^m x_i}{mx^{\mathrm{max}+}} \\
&= f(\mathbf{x}).
\end{aligned}
$$

This shows that the sparsity of $\mathbf{x}$ remains the same even when it is concatenated with itself $\delta$ times. □

**Proof of Bill Gates.** As one individual element becomes infinitely large, the sparsity increases. Formally, for every $i$, there exists $\rho > 0$, such that for all $\alpha > 0$:

$$f([x_1, ..., x_i + \rho + \alpha, ..., x_m]) > f([x_1, ..., x_i + \rho, ..., x_m])$$

The implication is that if a single value grows without bounds, the sparsity also increases indefinitely.

*Proof.* We first choose a sufficiently large $\rho$ such that $\forall i, x_i + \rho > x^{\max +}$.

Assume for contradiction that $S([x_1, ..., x_i + \rho + \alpha, ..., x_m]) \leq S([x_1, ..., x_i + \rho, ..., x_m])$.

This implies:

$$1 - \frac{\Sigma_{k=1}^m x_k + \rho + \alpha}{m(x_i + \rho + \alpha)} \leq 1 - \frac{\Sigma_{k=1}^m x_k + \rho}{m(x_i + \rho)}$$

$$\frac{\Sigma_{k=1}^m x_k + \rho + \alpha}{m(x_i + \rho + \alpha)} \geq \frac{\Sigma_{k=1}^m x_k + \rho}{m(x_i + \rho)}$$

$$\frac{\Sigma_{k \neq i} x_k + x_i + \rho + \alpha}{x_i + \rho + \alpha} \geq \frac{\Sigma_{k \neq i} x_k + x_i + \rho}{x_i + \rho}$$

$$\frac{\Sigma_{k \neq i} x_k}{x_i + \rho + \alpha} \geq \frac{\Sigma_{k \neq i} x_k}{x_i + \rho}$$

$$\frac{1}{x_i + \rho + \alpha} \not\geq \frac{1}{x_i + \rho},$$

leading to a contradiction. Therefore,

$$f([x_1, ..., x_i + \rho + \alpha, ..., x_m]) > f([x_1, ..., x_i + \rho, ..., x_m]).$$

$\square$

**Proof of Babies.** Adding a new element of zero increases sparsity. Formally, for a vector $\mathbf{x}$, the function $f$ satisfies:

$$f(\mathbf{x}||0) > f(\mathbf{x}).$$

Concatenating zero elements to the existing values increases the relative difference to the other values, which increases sparsity.

*Proof.*

$$1 - \frac{\Sigma_{k=1}^m x_k}{m+1} > 1 - \frac{\Sigma_{k=1}^m x_k}{m}$$

We start by subtracting one from both sides of the inequality:

$$-\frac{\Sigma_{k=1}^m x_k}{m+1} > -\frac{\Sigma_{k=1}^m x_k}{m}.$$

The negative sign can be removed by reversing the inequality:

$$\frac{\Sigma_{k=1}^m x_k}{m+1} < \frac{\Sigma_{k=1}^m x_k}{m}.$$

Multiplying both sides of the inequality by $m(m+1)$ (since $m$ is a positive integer, $m+1$ is also positive, and the inequality sign will not change), we get:

$$m\Sigma_{k=1}^m x_k < (m+1)\Sigma_{k=1}^m x_k.$$

Subtracting $m\Sigma_{k=1}^m x_k$ from both sides of the inequality, we obtain:

$$0 < \Sigma_{k=1}^m x_k.$$

Since $x_k$ is not a negative value, the sum is also non-negative, which verifies the inequality. Therefore, the original inequality is proven. $\square$

# D  Deciding Model Configurations for Investigation

## D.1  Investigation of Depth Impact on Accuracy in VGG16 Models

Table E.2: Experimental results of shortened VGG16 models with the Swap order for CIFAR-100. The number of removed convolution layers in the VGG16_n model is the difference between 16 and n.

|          | VGG16 | VGG16_15 | VGG16_14 | VGG16_13 | VGG16_12 | VGG16_11 | VGG16_10 | VGG16_9 | VGG16_8 |
|----------|-------|----------|----------|----------|----------|----------|----------|---------|---------|
| Accuracy | 70.93 | 72.97    | 73.63    | 73.77    | 74.00    | **74.11** | 72.31   | 70.95   | 70.76   |

Table E.3: Experimental comparison between the official VGG11 and our VGG16_11 models trained by CIFAR-100.

| Models | Order | |
|--------|------------|------|
|        | Convention | Swap |
| VGG11    | 64.55 | 69.94 |
| VGG16_11 | 69.5  | 74.11 |

To identify a model for focused analysis using the CIFAR dataset, we examined various VGG16 variants. This examination progressively removes convolution layers from the end towards the front of VGG16. The result can be seen in Table E.2. We observed an increase in accuracy until peaking at the VGG16_11 model, followed by a decline. Although a VGG11 model has already been proposed in Simonyan & Zisserman [18], the validation accuracy of VGG16_11 is significantly higher than VGG11 on CIFAR-100. The results can be seen in Table E.3.

# E  Training Hyperparameters

We sweep the learning rate and weight decay hyperparameter. The learning rate was 0.1 and 0.01. For CIFAR and Tiny-ImageNet datasets, we trained models with a batch size of 128, and the learning rate was reduced by one-tenth at 100 and 150 of the total 200 epochs, and we swept four weight decay of 0.005, 0.001, 0.0005, and 0.0001. For ImageNet datasets, we trained models with a batch size of 256, and the learning rate was reduced by one-tenth at 30 and 60 of the total 100 epochs, and we swept three weight decays of 0.001, 0.0005, and 0.0001. We chose the best averaged accuracy model for the three random seeds and averaged the values of these three models for all measurements for analysis. Because of the computation issue, we only use one seed for the ImageNet dataset with early stopping. The hyperparameters of VGG, MobileNet, and PreAct-ResNet are shown in E.1, E.2, E.3, respectively.

## E.1 VGG

Table F.1: Hyperparameters of VGG11 with Tanh

|  | Convention | Swap | NWDBN |
|---|---|---|---|
| Training Epochs | 200 | 200 | 200 |
| Learning Rate | 0.1 | 0.1 | 0.1 |
| Learning Rate Drop | 100, 150 | 100, 150 | 100, 150 |
| Weight Decay | 0.0005 | 0.0005 | 0.0005 |
| Batch Size | 128 | 128 | 128 |

Table F.2: Hyperparameters of VGG16 with Tanh

|  | Convention | | | | Swap | | | |
|---|---|---|---|---|---|---|---|---|
|  | CIFAR-10 | CIFAR-100 | Tiny ImageNet | ImageNet | CIFAR-10 | CIFAR-100 | Tiny ImageNet | ImageNet |
| Training Epochs | 200 | 200 | 200 | 90 | 200 | 200 | 200 | 90 |
| Learning Rate | 0.1 | 0.01 | 0.01 | 0.01 | 0.01 | 0.01 | 0.01 | 0.01 |
| Learning Rate Drop | 100, 150 | 100, 150 | 100, 150 | 30, 60 | 100, 150 | 100, 150 | 100, 150 | 30, 60 |
| Weight Decay | 0.0001 | 0.0005 | 0.001 | 0.0001 | 0.001 | 0.0005 | 0.001 | 0.001 |
| Batch Size | 128 | 128 | 128 | 256 | 128 | 128 | 128 | 256 |

Table F.3: Hyperparameters of VGG16 with the shifted Tanh

|  | Convention | | | | Swap | | | |
|---|---|---|---|---|---|---|---|---|
|  | CIFAR-10 | CIFAR-100 | Tiny ImageNet | ImageNet | CIFAR-10 | CIFAR-100 | Tiny ImageNet | ImageNet |
| Training Epochs | 200 | 200 | 200 | 90 | 200 | 200 | 200 | 90 |
| Learning Rate | 0.1 | 0.1 | 0.1 | 0.1 | 0.1 | 0.1 | 0.1 | 0.1 |
| Learning Rate Drop | 100, 150 | 100, 150 | 100, 150 | 30, 60 | 100, 150 | 100, 150 | 100, 150 | 30, 60 |
| Weight Decay | 0.0001 | 0.0005 | 0.0001 | 0.0001 | 0.0005 | 0.0005 | 0.0005 | 0.0001 |
| Batch Size | 128 | 128 | 128 | 256 | 128 | 128 | 128 | 256 |

Table F.4: Hyperparameters of VGG16 with the ReLU

|  | Convention | | | | Swap | | | |
|---|---|---|---|---|---|---|---|---|
|  | CIFAR-10 | CIFAR-100 | Tiny ImageNet | ImageNet | CIFAR-10 | CIFAR-100 | Tiny ImageNet | ImageNet |
| Training Epochs | 200 | 200 | 200 | 90 | 200 | 200 | 200 | 90 |
| Learning Rate | 0.01 | 0.01 | 0.1 | 0.1 | 0.01 | 0.01 | 0.01 | 0.01 |
| Learning Rate Drop | 100, 150 | 100, 150 | 100, 150 | 30, 60 | 100, 150 | 100, 150 | 100, 150 | 30, 60 |
| Weight Decay | 0.001 | 0.005 | 0.0001 | 0.0001 | 0.001 | 0.005 | 0.001 | 0.0005 |
| Batch Size | 128 | 128 | 128 | 256 | 128 | 128 | 128 | 256 |

## E.2 MobileNet

Table F.5: Hyperparameters for MobileNet with Tanh

| | Convention | | | | Swap | | | |
|---|---|---|---|---|---|---|---|---|
| | **CIFAR-10** | **CIFAR-100** | **Tiny ImageNet** | **ImageNet** | **CIFAR-10** | **CIFAR-100** | **Tiny ImageNet** | **ImageNet** |
| Training Epochs | 200 | 200 | 200 | 90 | 200 | 200 | 200 | 90 |
| Learning Rate | 0.1 | 0.1 | 0.01 | 0.1 | 0.1 | 0.1 | 0.01 | 0.1 |
| Learning Rate Drop | 100, 150 | 100, 150 | 100, 150 | 30, 60 | 100, 150 | 100, 150 | 100, 150 | 30, 60 |
| Weight Decay | 0.0001 | 0.0005 | 0.005 | 0.0001 | 0.0001 | 0.0005 | 0.005 | 0.0001 |
| Batch Size | 128 | 128 | 128 | 256 | 128 | 128 | 128 | 256 |

Table F.6: Hyperparameters for MobileNet with the Shifted Tanh

| | Convention | | | | Swap | | | |
|---|---|---|---|---|---|---|---|---|
| | **CIFAR-10** | **CIFAR-100** | **Tiny ImageNet** | **ImageNet** | **CIFAR-10** | **CIFAR-100** | **Tiny ImageNet** | **ImageNet** |
| Training Epochs | 200 | 200 | 200 | 90 | 200 | 200 | 200 | 90 |
| Learning Rate | 0.1 | 0.1 | 0.01 | 0.1 | 0.1 | 0.1 | 0.01 | 0.01 |
| Learning Rate Drop | 100, 150 | 100, 150 | 100, 150 | 30, 60 | 100, 150 | 100, 150 | 100, 150 | 30, 60 |
| Weight Decay | 0.0001 | 0.0005 | 0.005 | 0.0001 | 0.0005 | 0.0005 | 0.005 | 0.0005 |
| Batch Size | 128 | 128 | 128 | 256 | 128 | 128 | 128 | 256 |

Table F.7: Hyperparameters for MobileNet with ReLU

| | Convention | | | | Swap | | | |
|---|---|---|---|---|---|---|---|---|
| | **CIFAR-10** | **CIFAR-100** | **Tiny ImageNet** | **ImageNet** | **CIFAR-10** | **CIFAR-100** | **Tiny ImageNet** | **ImageNet** |
| Training Epochs | 200 | 200 | 200 | 90 | 200 | 200 | 200 | 90 |
| Learning Rate | 0.01 | 0.01 | 0.1 | 0.01 | 0.01 | 0.01 | 0.1 | 0.1 |
| Learning Rate Drop | 100, 150 | 100, 150 | 100, 150 | 30, 60 | 100, 150 | 100, 150 | 100, 150 | 30, 60 |
| Weight Decay | 0.001 | 0.005 | 0.0001 | 0.0001 | 0.001 | 0.005 | 0.001 | 0.0001 |
| Batch Size | 128 | 128 | 128 | 256 | 128 | 128 | 128 | 256 |

## E.3 PreAct-ResNet

Table F.8: Hyperparameters for PreAct-ResNet18 and PreAct-ResNet34 with Tanh. PreAct-ResNet18 is for the CIFAR dataset, and PreAct-ResNet34 is for the Tiny ImageNet dataset.

|  | Convention | | | Swap | | |
|---|---|---|---|---|---|---|
|  | CIFAR-10 | CIFAR-100 | Tiny ImageNet | CIFAR-10 | CIFAR-100 | Tiny ImageNet |
| Training Epochs | 200 | 200 | 200 | 200 | 200 | 200 |
| Learning Rate | 0.1 | 0.1 | 0.1 | 0.1 | 0.1 | 0.1 |
| Learning Rate Drop | 100, 150 | 100, 150 | 100, 150 | 100, 150 | 100, 150 | 100, 150 |
| Weight Decay | 0.0001 | 0.0001 | 0.0001 | 0.0005 | 0.0005 | 0.0005 |
| Batch Size | 128 | 128 | 128 | 128 | 128 | 128 |

Table F.9: Hyperparameters for PreAct-ResNet18 and PreAct-ResNet34 with the Shifted Tanh. PreAct-ResNet18 is for the CIFAR dataset, and PreAct-ResNet34 is for the Tiny ImageNet dataset.

|  | Convention | | | Swap | | |
|---|---|---|---|---|---|---|
|  | CIFAR-10 | CIFAR-100 | Tiny ImageNet | CIFAR-10 | CIFAR-100 | Tiny ImageNet |
| Training Epochs | 200 | 200 | 200 | 200 | 200 | 200 |
| Learning Rate | 0.1 | 0.1 | 0.01 | 0.1 | 0.1 | 0.1 |
| Learning Rate Drop | 100, 150 | 100, 150 | 100, 150 | 100, 150 | 100, 150 | 100, 150 |
| Weight Decay | 0.0001 | 0.0005 | 0.001 | 0.0005 | 0.0005 | 0.0005 |
| Batch Size | 128 | 128 | 128 | 128 | 128 | 128 |

Table F.10: Hyperparameters for PreAct-ResNet18 and PreAct-ResNet34 with ReLU. PreAct-ResNet18 is for the CIFAR dataset, and PreAct-ResNet34 is for the Tiny ImageNet dataset.

|  | Convention | | | Swap | | |
|---|---|---|---|---|---|---|
|  | CIFAR-10 | CIFAR-100 | Tiny ImageNet | CIFAR-10 | CIFAR-100 | Tiny ImageNet |
| Training Epochs | 200 | 200 | 200 | 200 | 200 | 200 |
| Learning Rate | 0.01 | 0.01 | 0.1 | 0.01 | 0.01 | 0.1 |
| Learning Rate Drop | 100, 150 | 100, 150 | 100, 150 | 100, 150 | 100, 150 | 100, 150 |
| Weight Decay | 0.005 | 0.005 | 0.0005 | 0.005 | 0.005 | 0.0005 |
| Batch Size | 128 | 128 | 128 | 128 | 128 | 128 |

