# OpenReview forum: "Tanh Works Better with Asymmetry"
_NeurIPS.cc/2023/Conference — NeurIPS 2023 poster_

### Official Review · Reviewer_hcmY · 2023-07-05

**Soundness:** 2 fair
**Presentation:** 2 fair
**Contribution:** 1 poor
**Rating:** 4
**Confidence:** 4

**Summary:**

This paper proves the hypothesis that asymmetric saturation benefits network performance by swapping the position of Batch Normalization and Tanh activation functions. The Swap model generates high sparsity and asymmetric saturation which enables Tanh to behave like a one-sided activation function. Experimental results show the asymmetric distributions consistently outperforms the symmetric ones. However, because the BN and Tanh combination hardly appears in modern networks and there is no convincing theory facilitating network architecture design, the contribution of this paper is of less significance.

**Strengths:**

1) This paper is easy to follow.
2) This paper provides a comprehensive experimental demonstration to validate asymmetric activation functions are superior to symmetric ones. This implies that ReLU-like activation functions are better than tanh activation functions.


**Weaknesses:**

1) One major concern is the lack of prevalence of the BN and Tanh combination in modern networks. As a result, the analysis and experiments conducted in this study hold limited empirical significance.
2) The benefits of asymmetric activation functions have previously been demonstrated from the perspectives of gradient [1] and expressivity [2]. However, this paper fails to contribute new explanations or adequately discuss the limitations of existing works.
3) The experiments in this paper are restricted to a limited range of network architectures. For example, even the widely used ResNet18 and ResNet50 models have not been evaluated on the ImageNet dataset.
4) A noticeable error can be identified in Figure 1, where the right bottom subplot indicates that 'BN' exhibits lower sparsity compared to 'Tanh,' contradicting the accompanying label stating 'High Sparsity.'

[1] Maas, A. L., Hannun, A. Y., & Ng, A. Y. Rectifier nonlinearities improve neural network acoustic models. 2013.
[2] Hanin, Boris, and David Rolnick. Complexity of linear regions in deep networks. 2019.


**Questions:**

1）The paper discusses sparsity and asymmetry, which have distinct mathematical definitions. However, it remains unclear which factor is the key determinant of network performance.
2） Whether the proposed swap model effectively addresses the issue of vanishing gradients?


**Limitations:**

The authors did not discuss Limitations in the paper.

---

> ### Author Rebuttal · Authors · 2023-08-09
>
> Thank you for your positive feedback and valuable suggestions.
>
> **Weakness 1.**
> Our main contribution lies beyond the combination of BN and Tanh. We have delved into understanding the significance of asymmetry in the activation functions, using Tanh as a base. This analysis can offer insights for future research on activation function design and interpretation.
>
> The derived insights can be applied to other functions, like ELU, which shares structural similarities with Tanh. ELU has the asymptote away from the x-axis. Thus, when ELU is used in the Convention order, its ability to generate asymmetry is hindered due to BN's zero-centered output. This imposed limitation on asymmetry drops the model accuracy. The accuracy of VGG16_11 trained by CIFAR-100 with ELU and other activation functions (Tanh, ReLU, leaky ReLU) can be seen in Table 1 in the PDF.
>
> Notably, Tanh and ELU outperform in the Swap order compared to the Convention order. This remarks our observation that restricted asymmetry degrades the accuracy. This enhanced performance can be seen in different models, not just VGG. Table 2 in the PDF file shows the accuracy of the ELU model in various settings.
>
> Moreover, shifted Tanh remains effective even in non-BN settings. Examining Convention and Swap orders, we discerned the importance of asymmetry and sparsity. Such understandings hold potential for performance enhancements, regardless of BN's presence. Notably, in the BN-free VGG16_11 model, shifted Tanh outperforms the Tanh, with an accuracy of 92.27% against Tanh's 89.68%.
>
> **Weakness 2.**
> As far as we know, no previous works discuss the advantages of asymmetry of the activation function.
> Our work differs in a critical aspect: asymmetric saturation. Specifically:
>  - [1] investigates gradient flow in the linear region of the ReLU in deep networks.
>  - [2] delves into the expressivity stemming from the linear regions formed by piecewise linear functions.
>
> These studies primarily focus on the linear aspects and not on asymmetric saturation.
> To illustrate further, [1] values the "sparse-dispersed code", where a few coding units are active at a given moment for an image, but all units contribute equally to coding over their lifetime. While this shared the sparsity perspective with ours, it is distinct from our main finding: asymmetric saturation helps improve performance.
>
> Thus, our paper sheds light on the benefits of asymmetric saturation, offering a novel perspective on the activation function.
>
> **Weakness 3.**
> We recognize the widespread use of ResNet architectures. However, they aren't suitable for our study, which focuses on analyzing the performance difference in the layer order.
>
> In the specific residual blocks of ResNet, BN exists in the skip connection. Thus, the number of BN and activation functions are different. This results in the Swap model having as many activation functions as BNs in the skip-connection, complicating a direct comparison with the Convention model.
>
> Nevertheless, the results of the ResNet model with Tanh on ImageNet and CIfar-100 are below. The ResNet-18 was conducted at one hyper-parameter (lr 0.1, wd 0.0001) due to limited time, and ResNet-20 for the CIFAR-100 is the best hyper-parameter accuracy.
>
> |              | Convention |   Swap   |
> |:------------:|:----------:|:--------:|
> |  ResNet-18 (ImageNet)   |    63.08   |   69.96  |
> |ResNet-20 (CIFAR-100)|    68.97   |   69.06  |
>
> Swap shows improved performance compared to Convention.
>
> Additionally, we extended our evaluation to PreAct-ResNet50 with Tanh on ImageNet. PreAct-ResNet, different from ResNet, lacks BN in its skip connection. This means that when the Swap order is employed, it does not introduce the challenges seen in ResNet. Our results with PreAct-ResNet50, at the best hyper-parameter. (Convention: 62.95, Swap: 72.82) The Swap model also outperforms the Convention model.
>
> **Weakness 4.**
> The sparse distribution refers to a state in which a small number of samples have large values.
> The right bottom subplot shows a state where most have a value of 0 and a small number of samples have a large value (e.g., -5).
> The averaged Sparsity over layers on Convention and Swap for VGG11 is as follows. (Convention: 0.718, Swap: 0.849)
>
> The lines in the subplot are thin and do not seem visible.
> We will update a figure that can add the existence range of values later.
>
> **Question 1.**
> It is not easy to control asymmetry and sparsity completely independently. However, we assume that asymmetry is more important in general. This can be confirmed by comparing the asymmetry and sparsity between the NWDBN and Convention models.
>
> When using NWDBN, it was not shown because the sparsity metric was not introduced. While Convention shows an average layer Skewness of 0.254 and Sparsity of 0.718, NWDBN presents a pronounced Skewness of 0.718 and a lower Sparsity of 0.288. Even with reduced Sparsity, NWDBN surpasses Convention as asymmetry increases.
>
> **Question 2.**
> The vanishing gradients problem is inevitable when using Tanh with excessive saturation. However, BN in the Swap order somewhat alleviates the vanishing gradients caused by asymmetric saturation in two aspects.
>
> 1. the scaling effect on the gradient by normalization, specifically $\sigma$ in BN. When asymmetric saturation occurs in Tanh, the $\sigma$ becomes smaller. The normalization in the forward pass is $(x-\mu)/\sigma=z$. And it becomes $dL/dx=1/\sigma*dL/dz$, which has the effect of scaling due to the $1/\sigma$.
>
> 2. the unboundedness of the convolution input. It can calculate a large weight gradient. In the Swap model, the convolution layer gets its input from BN, unlike the Convention model, which uses Tanh's output. When calculating the weight gradient of convolution weight, unbounded large input can create a larger gradient than Convention.
>
> In this respect, the Swap model can exhibits performs well across various settings even with asymmetric saturation.

---

> > ### Comment · Reviewer_hcmY · 2023-08-18
> >
> > Thanks for the careful response. I increase my rating to bordline reject considering that my concerns are partially addressed. I acknowledge the design of swapping the order of tanh and BN but still have concerns on the main insight provided by this paper that asymmetric saturation helps improve performance is significant. As recognized by the authors, asymmetry is highly associated with sparsity. It is commonly accepted that sparsity is the key to achieve a good performance (references like [5] in text). The benefits from asymmetry could be explained by sparsity under the intuition that sparse activations filter out some feature values at each layer and help distinguish in-class data and out-class data. In this way, the insight about asymmetric activation is less important. To emphasize asymmetry, the authors need to provide new findings on top of existing explanations on sparsity. I also suggest the authors to introduce the definition of sparsity metric at the first of paper since it is different from the usual definition referring to the number of non-zero values.

---

> > > ### Author Response · Authors · 2023-08-21
> > > **Response to Reviewer hcmY**
> > >
> > > Thank you for your thoughtful response and for raising your score.
> > >
> > > We do agree that sparsity and asymmetry often appear coincidently. However, the NWDBN model shows accuracy improved with increased asymmetry despite decreased sparsity. This observation holds for the LeCun Tanh and Softsign activation functions as well. Our findings suggest that asymmetry itself can boost accuracy, even if sparsity is not induced.
> > >
> > > We compared the Convention and NWDBN of VGG16_11 with the Softsign and LeCun Tanh activation functions. All measurements were obtained using the optimal hyper-parameters, and the results were averaged across three different seeds. Please see the below Table for details on accuracy, average Skewness, and average Sparsity.
> > >
> > > |                          | Convention with SoftSign | NWDBN with SoftSign | Convention with LeCun Tanh | NWDBN with  LeCun Tanh |
> > > |:-----------------:|:-----------------------------:|:--------------------------:|:--------------------------------:|:-------------------------------:|
> > > | accuracy          |                              70.01   |                            72.26  |                                   67.82   |                                  72.04  |
> > > | avg. Skewness |                             0.407   |                             1.393  |                                   0.430   |                                  1.377  |
> > > | avg. Sparsity    |                             0.664   |                             0.385  |                                   0.623   |                                  0.354  |
> > >
> > > Results show that NWDBN models with higher asymmetry and lower sparsity outperform the conventional models. These results support that asymmetric saturation plays a crucial role in enhancing accuracy.
> > >
> > > Regarding your comment on the definition of sparsity, we will introduce the sparsity definition earlier in the final submission, as you suggested.

---

### Official Review · Reviewer_s3KQ · 2023-07-05

**Soundness:** 3 good
**Presentation:** 4 excellent
**Contribution:** 2 fair
**Rating:** 6
**Confidence:** 3

**Summary:**

This paper investigates the performance of different activation function orders in deep learning models with batch normalization. The authors focus on the conventional order, where batch normalization is placed before the activation function, and the swapped order, where batch normalization is placed after the activation function. Surprisingly, they find that the swapped order achieves significantly better performance than the conventional order when using bounded activation functions like Tanh. The paper provides a thorough analysis of the underlying mechanisms and presents empirical evidence to support their findings.



**Strengths:**


- Novelty: The paper explores an interesting, and up to my knowledge previously overlooked aspect of activation function order in the context of batch normalization. The findings challenge the conventional wisdom and offer a new perspective on designing deep learning models.

- Comprehensive experiments: The authors conduct extensive experiments and carefully examine the output distributions of individual activation functions. Their investigation into the asymmetric saturation phenomenon provides lots of intuitions into the behavior of bounded activation functions that make intuitive sense.

- While focusing on Tanh as the primary activation function, the authors demonstrate that their findings are applicable to similar antisymmetric and bounded activation functions.

- Performance Improvement: The swapped order, combined with bounded activation functions and batch normalization, consistently outperforms the conventional order across various benchmarks and architectures. The results highlight the potential for achieving superior performance by exploiting the benefits of asymmetry and sparsity.

- Quality of presentation of ideas: the paper is very well written and gives the ideas in a clear and easy to follow manner. The contributions are clearly stated and there are no over claims in the text.


**Weaknesses:**

*Limited scope*
My main concern with this paper is that the limited scope of the network configurations considered for the experiments. For example, the results presented in Table 1 shows that conventional models with ReLU are mostly equal or better than all swap models, as well as the ReLU activation with swap order. Can authors make any further comments on possible reasons for this? For example, one might discount the strength of the main empirical evidence that is presented (differential performance between swapped & conventional models for Tanh & Tanh shifted), is only observed for the sub-optimal models and not the original model. While this does not directly contradict the main message of the paper, it significantly weakens its potential impact and scope. If the authors can address these questions by further experiments this could potentially strengthen the main conclusions made in the paper.

On a similar note, given the central hypothesis that asserts "The experiments designed to induce a different degree of asymmetric saturation support the hypothesis that asymmetric saturation helps improve performance, " can this hypothesis explain the differential performance between ReLU & Tanh too? Can this hypothesis explain the difference & benefit of having BN layers at all? (if you remove them, you don't have the asymmetric saturation ? There seems to be plenty of adjacent configurations that can be added to expand on the generalizability of this central idea of the paper.  These types of additional experiments could strengthen the study's conclusions.

*Lack of Theoretical Analysis* Although the paper presents compelling empirical evidence, a deeper theoretical analysis would enhance the understanding of why and how the swapped order, asymmetry, and sparsity contribute to performance improvement. Incorporating theoretical insights could strengthen the paper's contribution to the field.

*Figures & tables:* One minor but important issue: The tables and figures currently lack confidence intervals or standard deviations





**Questions:**

no questions

**Limitations:**

yes

---

> ### Author Rebuttal · Authors · 2023-08-09
>
> Thank you for your positive feedback and valuable suggestions.
>
> **Weakness: Limited scope**
>
> - Possible reasons for the Convention ReLU performance
>
> As the original author of BN[1] suggested, Convention seems to be fundamentally better than Swap. The good performance of Convention ReLU derives from these advantages. However, in the case of Tanh, the asymmetry caused by the Swap overcomes the disappeared advantages of the BN in the Convention order. We can confirm that Convention is better in the shifted Tanh, which can produce asymmetry even without a Swap.
>
> - Differential performance between swapped & conventional models
>
> Would you be kind enough to clarify what you mean by "sub-optimal model"? For reference, the VGG used in Table 1 is not VGG11 but VGG16. Only VGG16 has only BN added to the original model in that table, while all other models remain in their original versions.
>
> - Explanation of the differential performance between ReLU & Tanh
>
> We believe it's possible. We assume that the primary reason Tanh underperforms compared to ReLU in general cases is due to asymmetry. Therefore, when we tested with Swap or shifted Tanh to induce asymmetry, we observed a performance similar to ReLU's. From this, we deduce that the reason for the lower performance of Tanh compared to ReLU was its asymmetry.
>
>  - The Difference and benefits of having BN
>
> There are various roles of BN, and there have been many studies on it. This paper focuses on analyzing the correlation between Tanh and BN rather than showing the general advantages of BN. Nevertheless, we assume that BN is needed to complement the low sparsity caused by the asymmetry of Tanh.
>
> In the case of Tanh, BN increases the sparsity of the block output in the Swap structure, which helps improve the performance of the model, especially when it is combined with asymmetry.
>
> The layer-wise Skewness and Sparsity of the VGG16_11 without BN, which is called the NoBN model, and the Swap model using Tanh can be seen in Figure 2 in the PDF file.
>
> In the absence of BN, increasing the weight size can indeed produce asymmetry. However, the degree of skewness observed isn't as prominent as that in the Swap model. We interpret this subdued asymmetry in the NoBN model as a result of reduced sparsity. This limited generation of strong asymmetry could potentially contribute to a decline in performance.
>
> On the other hand, in the Swap structure, even if a high asymmetry is generated due to the BN located behind, it can create a high sparsity.
>
> **Weakness: Figures & tables**
>
> We have added confidence intervals for Figure 1 to the PDF. We will update the standard deviation for Table 1 in the main paper.
>
> [1] Ioffe, Sergey, and Christian Szegedy. "Batch normalization: Accelerating deep network training by reducing internal covariate shift." International conference on machine learning. pmlr, 2015.

---

> > ### Comment · Reviewer_s3KQ · 2023-08-12
> >
> > I would like to thank the authors to respond to my queries and questions. I do find the answers to be satisfactory and thus have decided to increase my score to weak accept.

---

### Official Review · Reviewer_AMuf · 2023-07-07

**Soundness:** 3 good
**Presentation:** 3 good
**Contribution:** 2 fair
**Rating:** 6
**Confidence:** 4

**Summary:**

This paper investigates neural network classifiers with bounded activation functions. The authors first observe that swapping the batch norm and activation order improves performance with bounded activation functions. They then observe that asymmetric saturation and sparsity occurs in the swap model compared to convention, and show how it correlates with accuracy. They then propose a modified activation function that promotes asymmetric saturation and shows that even in convention order, it benefits performance.

**Strengths:**

- The paper notes an interesting observation regarding the swap model with bounded activation
- It supports the hypothesis that asymmetric saturation and sparsity benefit benefit. This is achieved by caring the weight decay on the batch norm layers.
- The comparable performance with the modified tanh activation with relu is quite interesting. I wonder if ReLU-like behavior is what is the best for performance or if there is room for improvement past this.

**Weaknesses:**

- The evaluation is somewhat limited, since it is only done on VGG on image classification tasks. It’s not clear if these results would also apply to different architectures, e.g. transformer-based ones, or on different tasks such as segmentation or text classification.
- Comparison with different ReLUs, such as LeakyRELU or ELU.

**Questions:**

See weaknesses.

---

> ### Author Rebuttal · Authors · 2023-08-09
>
> Thank you for your positive feedback and valuable suggestions.
>
> **Weakness 2**
> The table below presents the accuracy of VGG16 models trained on CIFAR-100. The accuracy of both leaky ReLU and ELU is the best hyperparameter, and the accuracies are averaged over three seeds.
>
> |              | Convention |   Swap   |
> |:------------:|:----------:|:--------:|
> |     ReLU     |    73.68   |   71.79  |
> |  leaky ReLU  |    74.75   |   73.61  |
> |      ELU     |    69.85   |   72.27  |
> |     Tanh     |    64.84   |   72.17  |
> | Shifted Tanh |    73.87   |   73.21  |
>
> The shifted Tanh model with Convention outperforms the others except for the leaky ReLU model with Convention.
>
> Interestingly, the ELU model, like Tanh, exhibits enhanced performance in the Swap order, reinforcing our assertion that "asymmetry enhances performance."
>
> Given ELU's asymptote away from the x-axis, its capability to establish asymmetry is constrained in the Convention order due to BN's zero-mean output. However, the Swap order can amplify its asymmetry, leading to performance boosts in multiple models, not just VGG.
>
> |              | Convention |   Swap   |
> |:------------:|:----------:|:--------:|
> |  MobileNet   |    68.72   |   70.26  |
> |PreAct-ResNet18|    74.64   |   75.6  |

---

> > ### Comment · Reviewer_AMuf · 2023-08-21
> >
> > I thank the authors for their response, and will keep my score as weak accept.

---

### Official Review · Reviewer_yn6T · 2023-07-10

**Soundness:** 2 fair
**Presentation:** 3 good
**Contribution:** 3 good
**Rating:** 7
**Confidence:** 3

**Summary:**

The paper investigate the order of Batch Normalization and activation functions, and founded bounded activation functions like Tanh works better in the swapped order unlike bounded one like ReLU. To explain this, the authors analyze the asymmetric saturation levels at both the layer and channel levels and find that the Swap model has higher saturation levels, especially in layers with excessive saturation. They also introduce a new model, NWDBN, that encourages asymmetric saturation and show that it improves accuracy compared to the convention model. The paper concludes that asymmetric saturation can help improve performance in neural networks.

**Strengths:**

- The paper revolve around the hypothesis that "asymmetric saturation helps improve performance", and analyize that ReLU and Tanh bring different asymmetric saturation levels, this sound interesting to me.
- The paper identify the high sparsity induced by Batch Normalization after bounded activation is functions and validate that the higher sparsity induced would booster the performance.

**Weaknesses:**

- The logic chain of the story in this paper need some improvement, see more in Questions part.
- The input of convolution layer in Figure 1 upper and lower part seem different, can the authoer explain why?
- Why is x-axis in Figure 1 lower-right BN figure of range (-5, 2.5), while I didn't see any value in the range of (-5, -1), this plot could be misleading.

**Questions:**

- The paper did not explain well why higher sparsity induced by asymmetric saturation would booster the performance.
- Is there a way to control Asymmetric Saturation to archieve trade-off between sparsity and performance?

**Limitations:**

As stated in Questions.

---

> ### Author Rebuttal · Authors · 2023-08-09
>
> Thank you for your positive feedback and valuable suggestions.
>
> **Weakness 1.**
>
> **Question 1.**
> The robustness of the noise obtained through the sparsity results in improved performance. Sparse representation indicates that relatively few units represent the data sample. Thus, even if the perturbation is added to the input, this noise less affects sparse features.
> The Swap model has strength on sparsity compared to the Convention model. In Section 3.3, we verified the robustness of the Swap model on the corrupted dataset. The increased sparsity of the Swap model outperforms the Convention model.
>
> **Question 2.**
> Thank you for the insightful question. However, it is not easy to completely isolate only asymmetry to achieve trade-offs between sparsity and performance.
>
> **Weakness 2.**
> For the Swap model, the mean of convolution layer output needs to shift substantially away from 0 to achieve asymmetry on Tanh. Consequently, this results in a distribution that is biased in one direction.
>
> Conversely, in the Convention model, there is no reason to shift the mean of convolution output. The normalization of BN adjusts the mean of the convolution layer output to zero. This normalization process can lead to the output of the convolution layer non-biased.
>
>
> **Weakness 3.**
> The subplot shows most samples near 0, with a few outliers like -5, remarking the Swap model's high sparsity. Due to the line's thinness, such values may be hard to discern.
> We will enhance Figure 1 for a more clear representation.

---

### Author Rebuttal · Authors · 2023-08-10

We thank the reviewers for their positive feedback and valuable suggestions.

---

### Decision · Program_Chairs · 2023-09-21

**Decision:**

Accept (poster)

**Comment:**

This paper studies different activation function orders the context of batch normalization. It investigates the setting where batch normalization is placed before the activation function, and the swapped order, and find empirically that the swapping leads to improved results for bounded activation functions, eg. tanh. The underlying mechanisms are analyzed, showing that asymmetric saturation benefits network performance by generating sparser activations. While this is theoretically interesting, one point of criticism from the reviewers is the limited practical significance of the work, because, the combination of BatchNorm and tanh is barely used in SotA models. Yet, reviewers and AC agree that the observation made is interesting. The authors are encouraged to include the additionally provided results on models beyond VGG into the camera ready version.